# Phenome-wide association study of loci harboring de novo tandem repeat mutations in UK Biobank exomes

Frank R. Wendt [1,2,3,4,5] ✉, Gita A. Pathak[4,5] & Renato Polimanti [4,5]

When present in coding regions, tandem repeats (TRs) may have large effects on protein structure and function contributing to health and disease. We use a family-based design to identify de novo TRs and assess their impact at the population level in 148,607 European ancestry participants from the UK Biobank. The 427 loci with de novo TR mutations are enriched for targets of microRNA-184 (21.1-fold, $P = 4.30 \times 10^{-5}$, FDR = $9.50 \times 10^{-3}$). There are 123 TR-phenotype associations with posterior probabilities > 0.95. These relate to body structure, cognition, and cardiovascular, metabolic, psychiatric, and respiratory outcomes. We report several loci with large likely causal effects on tissue microstructure, including the *FAN1*-[TG]$_N$ and carotid intima-media thickness (mean thickness: beta = 5.22, $P = 1.22 \times 10^{-6}$, FDR = 0.004; maximum thickness: beta = 6.44, $P = 1.12 \times 10^{-6}$, FDR = 0.004). Two exonic repeats *FNBP4*-[GGT]$_N$ and *BTN2A1*-[CCT]$_N$ alter protein structure. In this work, we contribute clear and testable hypotheses of dose-dependent TR implications linking genetic variation and protein structure with health and disease outcomes.

Tandem repetitive elements (TRs) are genomic loci consisting of consecutively repeated basepair motifs and represent one of the largest sources of genetic variation in humans[1]. TR motifs range from 1 to >20 basepairs and can be repeated dozens of times[2]. Variation in TR copy number has been associated with many diseases, including Huntington's disease, which is typically characterized by over 40 copies of a CAG motif in *HTT*[2]. In combination, the need for large DNA sequencing datasets and the complexity of aligning repetitive DNA sequence reads to reference genomes[3–6] have contributed to the omission of TRs from large genome-wide studies of health and disease[7,8]. Phenome-wide association studies of large TRs (i.e., TRs with >9 basepairs in the repeat unit) revealed likely causal effects on height, hair morphology, and several biomarkers of human health[8]. Most notably, these TRs have substantially larger, and independent, effects on phenotype relative to nearby SNPs detected by genome-wide association studies. To date, small TRs between 1–9 basepairs have not been investigated for phenotypic consequences in large cohorts but they hold considerable

promise for explaining a portion of missing heritability not captured in GWAS[9].

De novo mutations are a class of genetic variation where offspring harbor an allele absent in either parent. Single nucleotide de novo mutations are often rare, deleterious, and may disrupt gene function contributing to many genetic disorders[10]. With respect to complex traits, the cumulative burden of de novo mutations may play a large additive role in disease etiology[11]. Due to their repetitive nature, TRs have much higher mutation rates than single-nucleotide variants making de novo mutational events more common for this class of genomic variation[12]. De novo TRs have been associated with complex traits, like autism spectrum disorder, in family trios[13]. Often studied in the context of a specific condition, it is unclear how pervasive de novo TR mutations are across the genome and how they affect human health and disease.

In this study, we apply a two-staged analytic approach to UK Biobank (UKB) data to first identify de novo TR mutations using a family-based design. These sites of the genome detected in relatively

[1]Department of Anthropology, University of Toronto, Mississauga, ON, Canada. [2]Biostatistics Division, Dalla Lana School of Public Health, University of Toronto, Toronto, ON, Canada. [3]Forensic Science Program, University of Toronto, Mississauga, ON, Canada. [4]Department of Psychiatry, Yale School of Medicine, New Haven, CT, USA. [5]VA CT Healthcare System, West Haven, CT, USA. ✉e-mail: frank.wendt@utoronto.ca

small samples may represent mutational hotspots relevant for phenotypic variation[14]. Second, we use a population-based design to characterize the effect of variation in these TRs on 1844 human traits (Fig. 1). We report 426 TR associations with body structure, cognition, and cardiovascular, metabolic, psychiatric, and respiratory outcomes. Fine-mapping these loci identifies 41 TRs with a high probability of large causal effects on structural features of the carotid artery and thalamus radiation, highlighting the critical role of TR variation on health and disease outcomes.

## Results

### Family trios and de novo mutations

After IBD analysis and quality control, 40 trios were identified (39 EUR and 1 AFR). There were 17 male and 22 female offspring ($\chi^2 = 0.119$, df = 1, $P = 0.423$) with no difference in age between the sexes (male mean = $42.60 \pm 1.94$, female mean = $43.05 \pm 1.79$, $t = -9.23$, df = 35.30, $P = 0.362$). Due to sample size limitations in diverse ancestry trios, the genetic analyses focus only on the 39 trios of EUR descent (Supplementary Data 1).

Five families had no detectable de novo TR mutations after applying the quality thresholds for this study. Across 34 EUR families, there were 1031 de novo TR mutations (mean number of TRs per family = $31.2 \pm 18.5$; Supplementary Data 2). These 1031 mutations reflect 427 loci, 211 of which were mutated in >1 trio (Fig. 2). There was a significant trend towards TR expansions versus contractions ($\chi^2 = 5.46$, df = 1, $P = 0.019$). The mean contraction size was $1.41 \pm 0.98$ repeat units (maximum mutation was −6 repeat units at $HTT$-$[CAG]_N$) and the mean expansion size was $-1.34 \pm 0.69$ repeat units (maximum mutation was 12 repeat units at $DST$-$[CA]_N$). The most frequently mutated TR was $NOTCH4$-$[AGC]_N$, observed in 11/39 trios and mutated between 1 and 3 repeat units per trio.

### Locus annotation

We applied two methods to annotate the TR loci with de novo mutations. First, we tested whether each TR was associated with the expression of the positionally mapped TR-containing gene. A set of 352 approximately LD-independent TRs was selected by selecting a single TR in a 200-kb window, prioritizing the locus with the larger number of de novo mutations in this dataset. Using these 352 genes, we performed hypergeometric tests for gene set enrichment in FUMA and ShinyGo. After multiple testing correction (FDR 5% performed per gene set category), there were eight gene sets significantly enriched in FUMA and ShinyGO (Table S3). The largest enrichment was related to targets of MIR184 (21.1-fold enrichment, FUMA $P = 4.30 \times 10^{-5}$, SinyGO FDR-adjusted $P = 0.01$). The remaining enrichments include growth hormone pathway, cell morphogenesis involved in differentiation (GO:0000904), targets of microRNAs 515-1, 515-2, and 519E, targets of transcription factors SP1 and SOX9, genes downregulated in regulatory T-cells relative to conventional T-cells (GSE13306), and genes involved in epithelial-mesenchymal transition (MSigDB M5930, https://www.gsea-msigdb.org/gsea/msigdb/cards/HALLMARK_EPITHELIAL_MESENCHYMAL_TRANSITION.html).

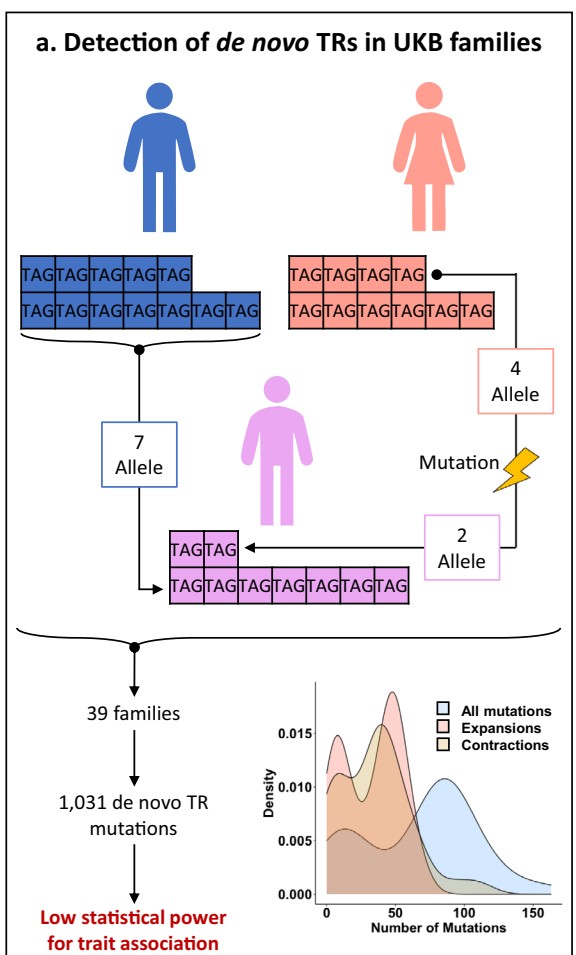

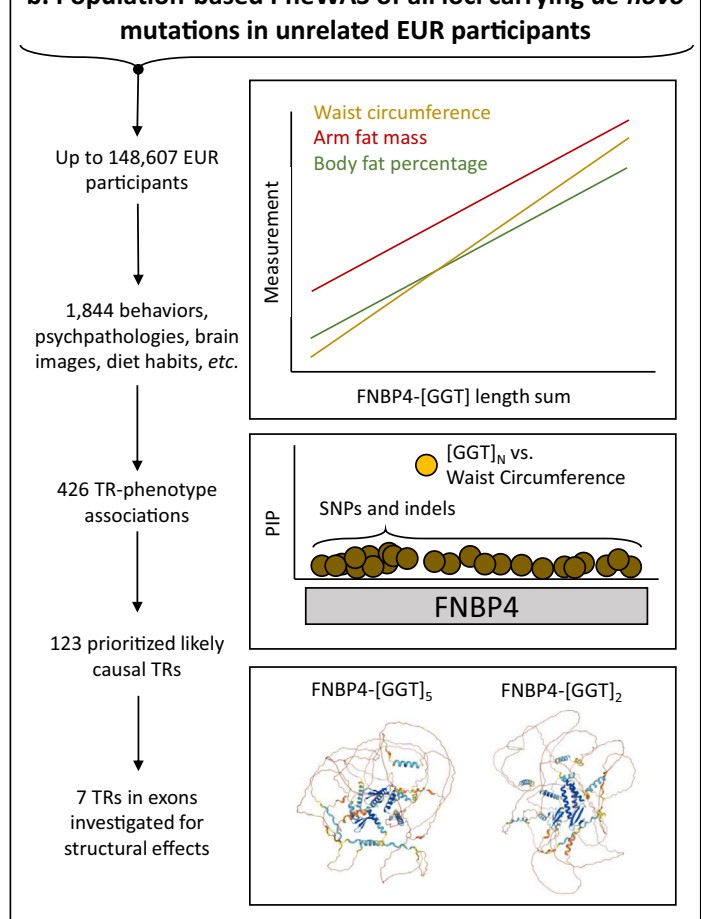

**Fig. 1 | Study overview and analysis plan.** (**a**) describes the family-based detection of de novo tandem repeats while (**b**) describes the population-based assessment of each locus. Abbreviations: tandem repeat (TR), phenome-wide association study (PheWAS), European ancestry (EUR), posterior inclusion probability (PIP), Formin Binding Protein 4 (*FNBP4*), single nucleotide polymorphism (SNP).

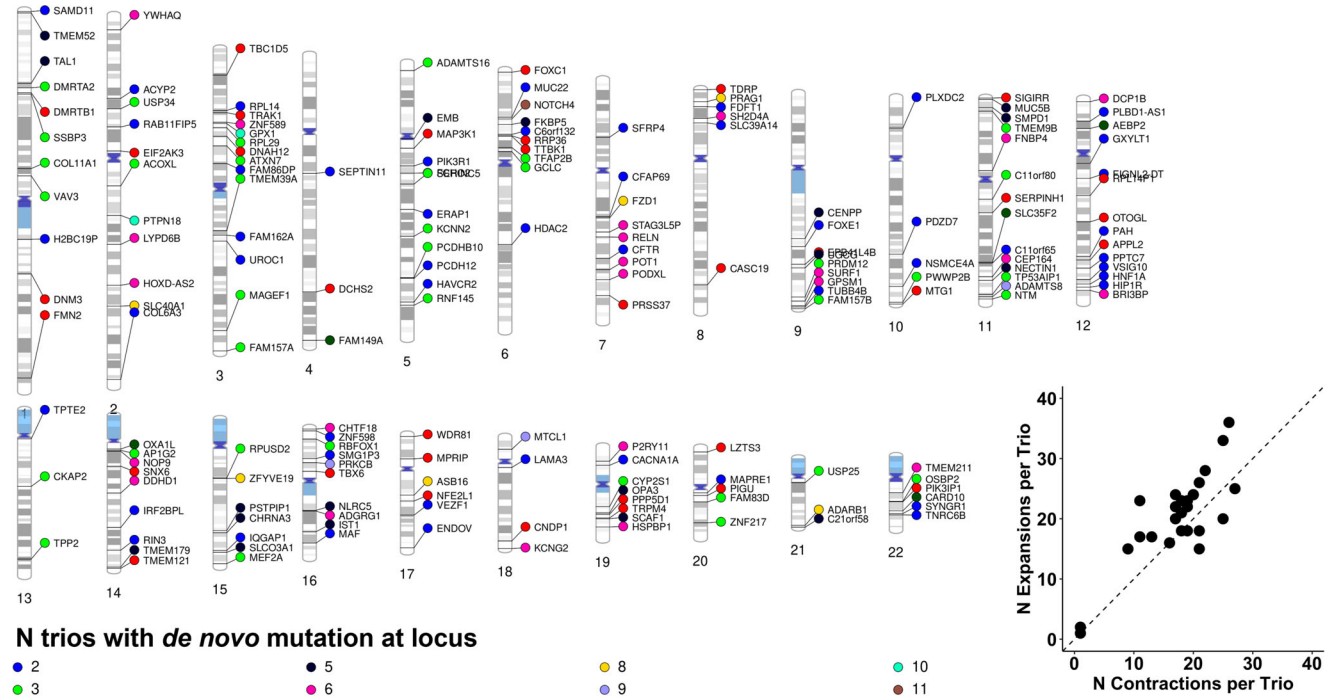

**Fig. 2 | De novo variants.** Chromosomal localization of de novo TR mutations observed in at least two trios. Each data point is a single TR mutation color coded to designate the number of UKB European ancestry trios in which that mutation was observed. Each data point is annotated with the gene containing the TR locus. Loci are positioned relative to the hg38 reference genome. The bottom right inset shows the slight bias in repeat expansions relative to repeat contractions per trio ($N = 39$). The diagonal dashed line indicates a 1:1 ratio of TR expansions:contractions.

Next, expression-associated TRs were analyzed with respect to 17 tissues in the GTEx repository[15]. From the same set of 352 approximately LD-independent TRs, there were 259 significant TR-gene expression associations across 17 tissues (Fig. 3). Among them, $CKAP2$-[GCGGTG]$_N$ was associated with expression across 17 tissues (Z-score range 3.26 in transformed fibroblasts to 11.77 in tibial artery).

## PheWAS of TRs harboring de novo mutations

Appreciating the low statistical power of the family-based design to identify phenomic attributes associated with each de novo mutation, we adopted a population-based PheWAS of each locus identified from the family-based design. The goal of this approach was to shed light on the phenotypic spectrum associated with TRs that may be hotspots for de novo mutational events.

We associated TR length sums with 1,844 phenotypes in up to 148,607 European ancestry participants. After removing TRs with low mean read depth (4 TRs) and high missingness rates (8 TRs), PheWAS was performed for 415 TRs. After FDR multiple testing correction ($P < 2.75 \times 10^{-5}$), there were 426 TR-phenotype associations, representing 97 TRs (Fig. 4 and Supplementary Data 5). The most significant and likely causal association was between $NCOA6$-[GT]$_N$ and *ease of skin tanning* (UKB Field ID 1727) such that individuals who burn rather than tan tend to have longer alleles at this locus (beta = 0.069, se = 0.003, $P = 1.51 \times 10^{-155}$, posterior probability = 1). The maximum Cohen's $d$ comparing effect sizes across *ease of skin tanning* categories was 0.212 (Get very tan *versus* Never tan, only burn $P = 9.21 \times 10^{-308}$; Fig. 5a and Supplementary Data 6).

For the 27 TRs with significant (FDR < 5%) associations across multiple trait domains, hypergeometric tests were applied to identify enriched domains (Supplementary Data 7). Twelve TRs were enriched for associations with hematological measures (mean fold-enrichment $57.9 \pm 55.8$), the most significant of which was $NOTCH4$-[AGC]$_N$

(64.5-fold enrichment of hematology associations, $P = 3.28 \times 10^{-16}$). The largest likely causal effect size observed for this locus was with respect to *total cholesterol* (UKB Field ID 30690) such that individuals with longer TR length have lower cholesterol concentration (beta = −0.014, se = 0.003, $P = 5.91 \times 10^{-6}$, posterior probability = 1). The maximum Cohen's $d$ comparing effect sizes across $NOTCH4$-[AGC]$_N$ length sum was 0.074 ($NOTCH4$-[AGC]$_{15}$ *versus* $NOTCH4$-[AGC]$_{24}$ $P = 2.25 \times 10^{-4}$; Fig. 5b and Supplementary Data 6).

## Notable likely causal TR-phenotype associations

After fine-mapping, there were 123 TR-phenotype associations involving 41 TRs with evidence that variation in the TR (posterior probability > 0.95), rather than a nearby SNP or indel, drove the genotype-phenotype association (Supplementary Data 8). Though not enriched among the de novo variants detected, there were notable relatively large effects and likely causal TRs associated with structural components of the white matter and peripheral vasculature. Variation in $BTN2A1$-[CCT]$_N$ was associated with increased isotropic volume fraction (ISOVF) in the right superior thalamic radiation (beta = $7.33 \times 10^{-4}$, se = $1.41 \times 10^{-4}$, $P = 2.19 \times 10^{-7}$; posterior probability = 0.989). The largest difference was observed between $BTN2A1$-[CCT]$_{12}$ and $BTN2A1$-[CCT]$_{16}$ (Cohen's $d = 0.209$; $P = 1.91 \times 10^{-163}$; Fig. 5c). Length variation at $FAN1$-[TG]$_N$ was associated with the mean (beta = 5.22, se = 1.08, $P = 1.22 \times 10^{-6}$; posterior probability = 0.979) and maximum (beta = 6.44, se = 1.32, $P = 1.12 \times 10^{-6}$; posterior probability = 0.985) carotid intima-medial thickness (IMT). For mean (Cohen's $d = 0.50$, $P = 0.013$) and maximum (Cohen's $d = 0.56$, $P = 0.016$) carotid intima-medial thickness, the largest effect size was observed between $FAN1$-[TG]$_{27}$ and $FAN1$-[TG]$_{44}$ (Fig. 5d).

We next revisited the family trios to test whether the top 10% of likely causal effect sizes could be detected in these samples. The relationship between platelet count (UKB Field ID 30080) and $USP30$-[CGG]$_N$ was recapitulated in probands (beta = −12.62, se = 4.99, $P = 0.017$) and had a significantly greater effect than in the general

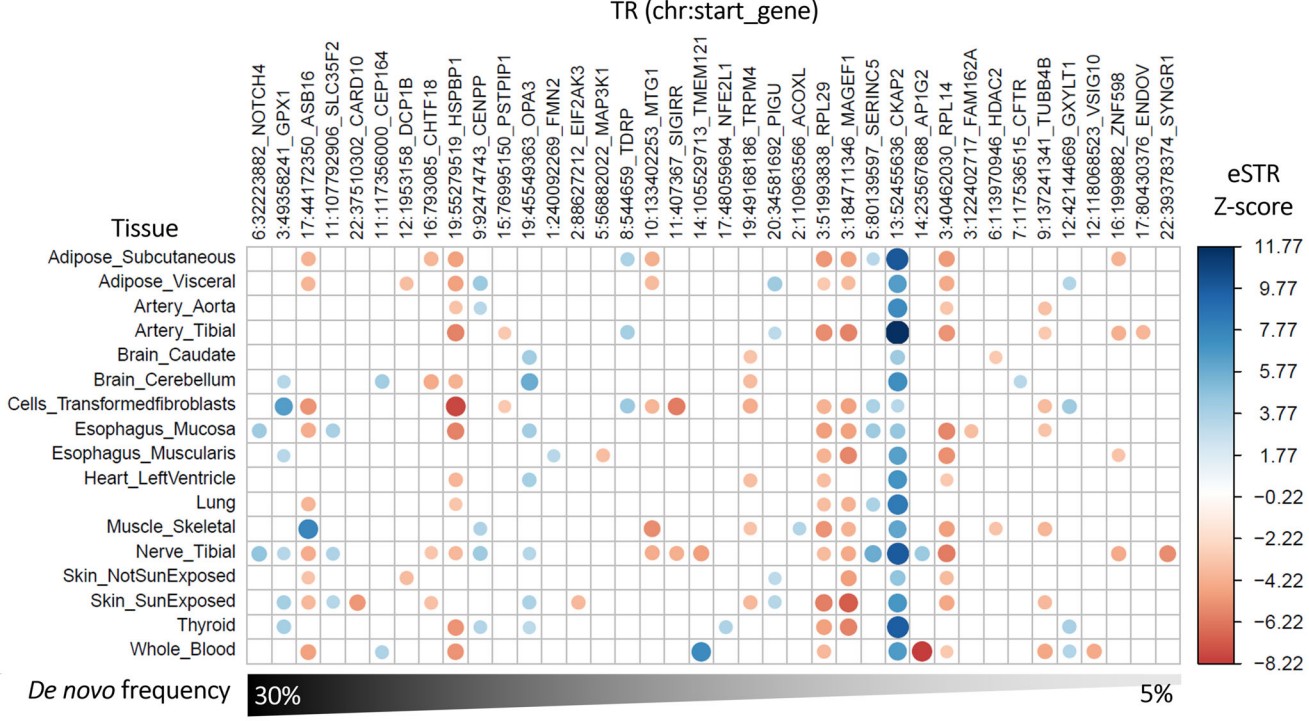

**Fig. 3 | De novo tandem repeats associated with gene expression (eSTR).**
Association between tandem repeats (TR) and gene expression across 17 tissues from GTEx[15]. Each TR is named according to its start position with respect to the hg38 reference genome and the gene containing the TR whose expression is influenced by variation at the TR. The heatmap is restricted to the TRs harboring de novo mutations in more than one trio (~5% frequency). All TR-gene-tissue association statistics are provided in Supplementary Data 4.

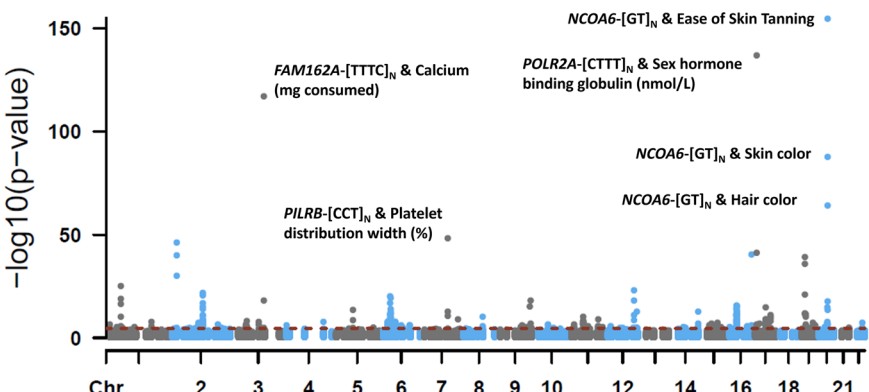

**Fig. 4 | TR-phenotype associations.** Manhattan plot of phenome-wide association results for 415 TRs and 1844 phenotypes in European ancestry participants of the UK Biobank. Linear or logistic regression was used to test to effect of each TR on linear/ordinal and binary traits, respectively. All models included age, sex, sex × age, age$^2$, sex × age$^2$, and the top 10 within-ancestry principal components as covariates. The dashed red line indicates the P-value threshold after multiple testing correction (FDR < 5%; $P = 2.75 \times 10^{-5}$). Select TR-phenotype associations are labeled. All summary statistics are provided in Supplementary Data 5. Chromosome (Chr), Family With Sequence Similarity 162 Member A (*FAM126A*), Paired Immunoglobin Like Type 2 Receptor Beta (*PILRB*), Nuclear Receptor Coactivator 6 (*NCOA6*), DNA-directed RNA polymerase II subunit RPB1 (*POLR2A*), milligrams (mg), nanomoles per liter (nmol/L).

population (beta = −1.39, se = 0.159, $P = 6.17 \times 10^{-19}$; $P_{\text{diff}} = 0.025$), though we caution readers to the potential for winner's curse when interpreting this observation.

## Other notable loci from the TR literature

The *HTT*-[CAG]$_N$ repeat expansion confers risk for Huntington's disease. None of the UKB trios investigated had *HTT*-[CAG]$_N$ alleles (size range from 9-17 CAG repeats Supplementary Data 2) that can be linked to Huntington's-like symptoms (CAG > 40). In our PheWAS, the *HTT*-[CAG]$_N$ repeat was associated with mean ISOVF in the right inferior cerebellar peduncle (beta = −7.25 × 10$^{-4}$, se = 1.51 × 10$^{-4}$, $P = 1.54 \times 10^{-6}$). The fine-mapping posterior probability for this association was 0.88,

which was lower than our desired confidence threshold 0.95. However, the posterior probabilities for bi-allelic variation in the surrounding region are substantially lower (range 1.54 × 10$^{-4}$ for rs231340 to 0.006 for rs755137).

## Putative structural effects of repetitive elements

Of the 41 genes with TRs exhibiting confident fine-mapping probabilities, the TRs implicated in these relationships were localized in introns (22/41), 5′ UTRs (7/41), 3′ UTRs (1/41), non-cytoplasmic exons (2/41), signal peptide binding regions (2/41), transmembrane helices (2/41), intrinsically disordered regions (1/41), and regions with unclear structure/function (4/41). Motivated by the understanding that protein

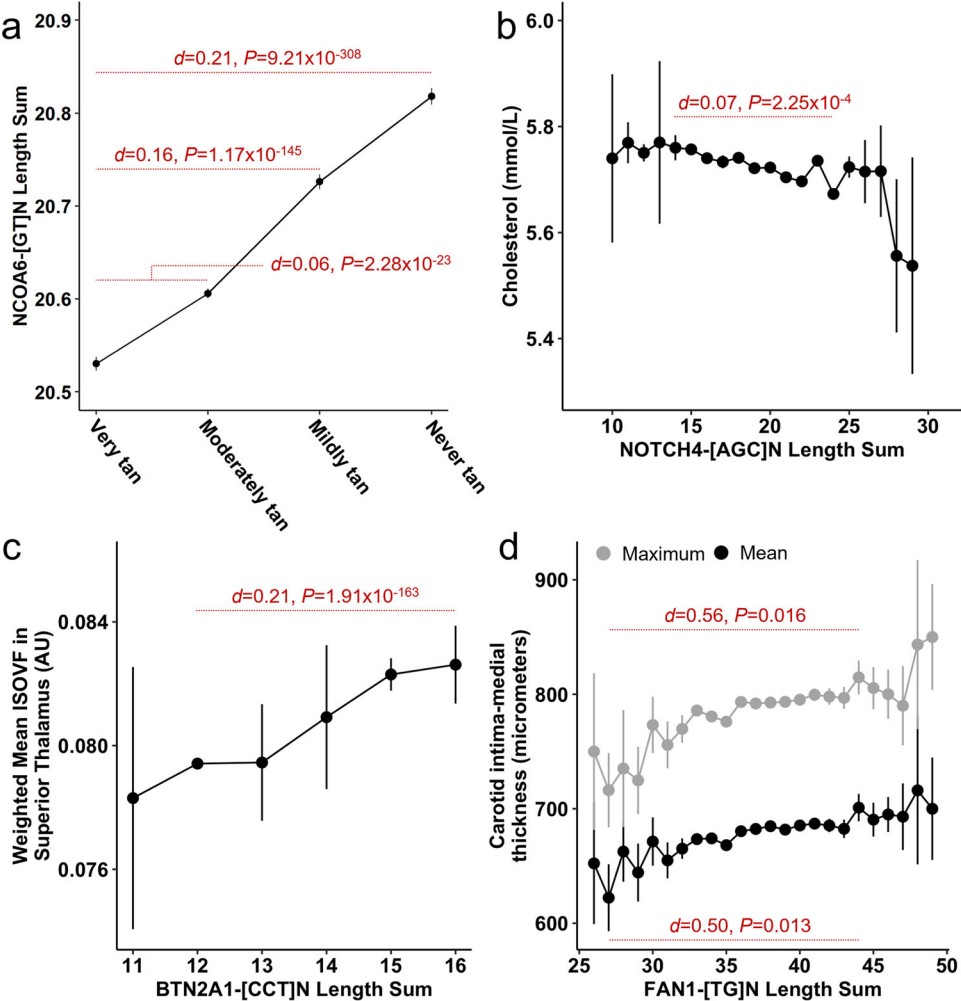

**Fig. 5 | Dose effects of TR length sums on select traits.** Effects of TR length burden on various phenotypes (Ease of skin tanning $N = 145,262$ participants in (**a**); Cholesterol concentration $N = 148,607$ participants in (**b**); ISOVF in tract superior thalamic radiation $N = 18,042$ participants in (**c**); carotid intima medial thickness $N = 21,736$ participants in (**d**)) in UK Biobank participants of European ancestry. Each data point represents the mean (±standard error) phenotype measurement at the indicated TR length. Red text denotes several small to medium effect size differences (Cohen's $d$ and two-sided $P$-value) between phenotype estimates. Nuclear Receptor Coactivator 6 (*NCOA6*), nanomoles per liter (nmol/L), Notch Receptor 4 (*NOTCH4*), isotropic volume fraction (ISOVF), arbitrary unit from magnetic resonance imaging (AU), Butyrophilin Subfamily 2 Member A1 (*BTN2A1*), FANCD2/FANCI-associated nuclease 1 (*FAN1*).

structure and function are intimately connected, we used AlphaFold[16] to compare the structure of canonical protein sequences and protein sequences that harbor expanded or contracted TRs. For seven genes (Supplementary Data 9), we evaluated protein structure using all TR allele lengths with frequencies >1%. To evaluate the effect of each TR allele length, we compared per-residue predicted local-distance difference test (pLDDT) scores between canonical and mutated proteins.

The *FNBP4*-[GGT]$_N$ contraction showed little effect on the three-dimensional structure of the residue-containing region (mean $\Delta$pLDDT = 3.41, se = 0.35). However, the presence of a contracted region of polar residues did associate with large increases in structure confidence of two proline-rich stretches (residues 709-728 mean $\Delta$pLDDT = 17.47, se = 2.06; residues 902−915 mean $\Delta$pLDDT = 21.32, se = 2.93; Fig. 6a, b). Furthermore, the FNBP4 binding pocket contains two tryptophan-rich regions that bind with proline-rich sequences of formins[17]. In the presence of *FNBP4*-[GGT]$_2$, the binding pocket beta sheet 1 (Supplementary Data 10) has a higher position error with each proline-rich region indicating worse structural stability relative to *FNBP4*-[GGT]$_5$ (proline rich region 1 difference = 1.52 Å, Cohen's $d = 1.22$, $P = 9.15 \times 10^{-86}$ and proline rich region 2 difference = 2.08 Å, Cohen's $d = 1.74$, $P < 9.21 \times 10^{-308}$; Fig. 6c).

The *BTN2A1*-[CCT]$_N$ expansion showed marked effects on the per-residue confidence of the containing leucine-rich alpha-helix relative to the canonical protein (mean $\Delta$pLDDT = −15.37, se = 4.03; Fig. 7a, b) suggesting superior stereochemical plausibility of the expanded form of the protein. This local change reduced the 3-dimensional structural alignment error of BTN2A1's Ig-like V-type and B30.2/SPRY regions by 3.40 Å (Cohen's $d = 0.918$, $P = 4.36 \times 10^{-100}$) and 0.638 Å (Cohen's $d = 1.32$, $P < 9.21 \times 10^{-308}$), respectively (Fig. 7c and Supplementary Data 12). Additional protein structural data are provided as Supplementary Information (Supplementary Figs. S1–S5).

## Discussion

TRs are a major source of genetic variation in humans that are under-investigated in common complex traits but have great potential to inform the missing heritability[9] and potential causal biology of disease states when localized to regulatory and coding regions of the genome. Due to their repetitive nature, TRs have relatively high mutation rates that enrich this class of variation for de novo mutations. Leveraging a two-staged approach using family-based and population-based designs, this study identified 427 loci harboring de novo TR mutations that may localize to hotspots of mutational events in the human genome[14]. There were 426 TR-phenotype associations across several

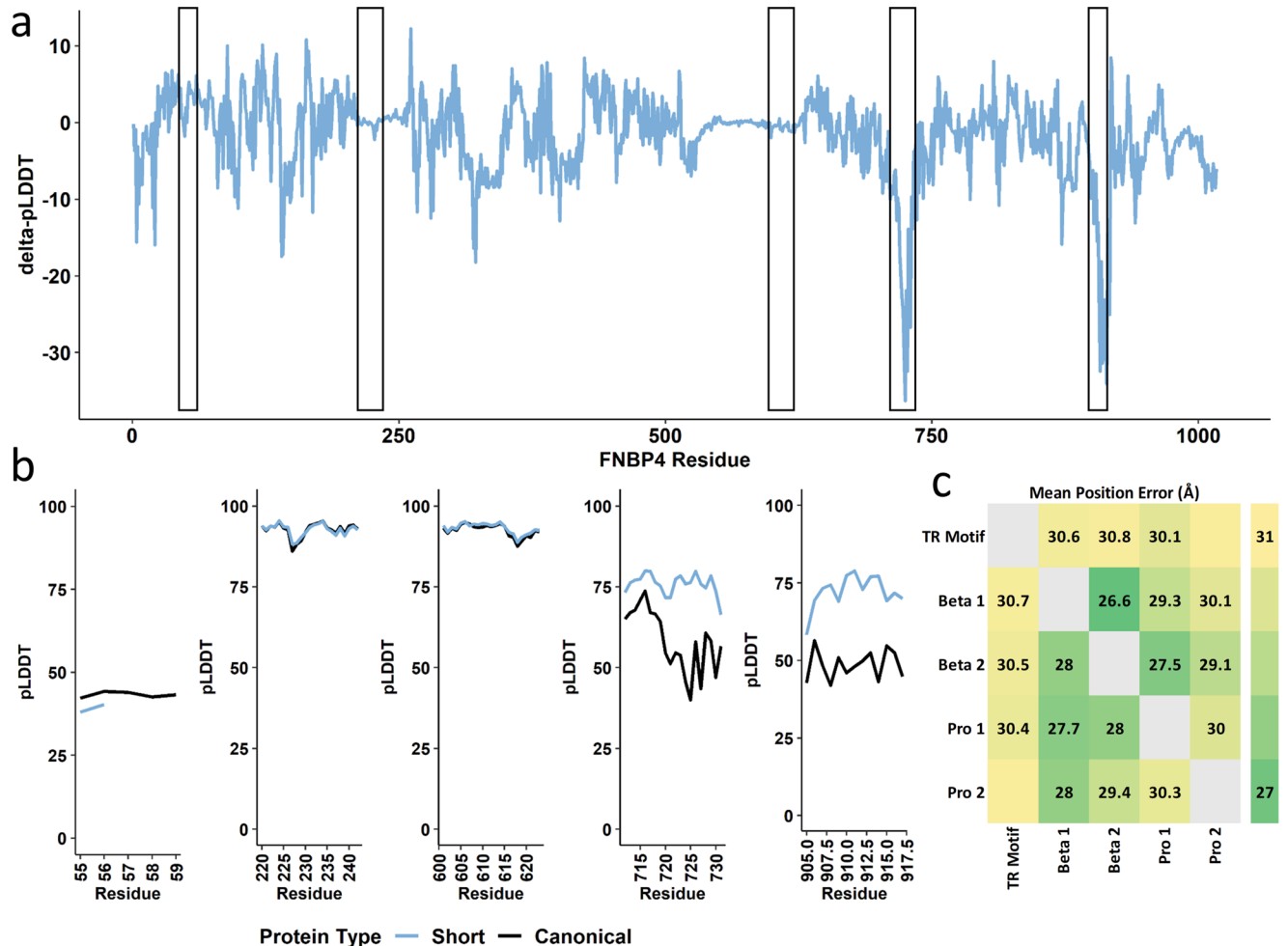

**Fig. 6 | Structural changes in Formin Binding Protein 4 (FNBP4) due to TR variation.** Putative impact of tandem repeat mutations on the structure of FNBP4. Change in per-residue folding confidence in canonical FNBP4 was compared to the short mutation (*FNBP4*-[GGT]$_2$) (**a**) and zoomed-in regional confidence estimates for both protein structures are shown in **b**. In **c**, the canonical (bottom left) and mutated (top right) predicted alignment errors are compared between the TR-containing motif, the two beta-sheet domains occupying FNBP4 active site, and two proline-rich stretches on the periphery of FNBP4's 3-dimensional structure that resemble formin. Cells occupied by text indicate a pairwise significant difference in angstrom (Å) between canonical and mutated FNBP4. Additional details for **c** are provided in Supplementary Data 11. Proline-rich sequence (Pro), per-residue predicted local-distance difference test (pLDDT).

health and disease domains with significant enrichment of loci associated with biomarkers such as cholesterol and alkaline phosphatase concentrations. Notably, there were 124 TR-phenotype associations where the TR, not surrounding SNPs or indels, was likely causal for trait variation thereby highlighting a large contribution to the genetic etiology of health and disease that is independent of signals identified by GWAS.

In a sample of European ancestry family trios from the UKB, we identified 427 loci harboring de novo TR mutations. The most frequently mutated TR mapped to *NOTCH4*. *NOTCH4* encodes neurogenic locus notch homolog 4 and is almost exclusively expressed in the vasculature where changes in gene expression produce less vasculature branching and reduced vessel integrity. NOTCH4 has been the subject of many studies of schizophrenia. However, the TR locus identified here is contested as a causal variant in the gene[18]. *NOTCH4*-[AGC]$_N$ variation was enriched for association with biomarkers and metabolic health, consistent with its localization to the vasculature. The strongest result identified longer copies of the *NOTCH4*-[AGC]$_N$ TR associated with lower total cholesterol concentration. The polyleucine stretch encoded by this TR makes up a portion of the NOTCH4 extracellular signal peptide domain and therefore may contribute to abnormal astrocyte differentiation, angiogenesis, and coronary vessel development[19].

As a whole, the collection of genes harboring mutated TRs was enriched for targets of the transcriptionally relevant micro-RNAs 184 and 5155P-519E and transcription factors Sp1 and SOX9. Micro-RNA 184 is a tumor suppressor implicated in numerous cancers as it plays a critical role in cell differentiation and fate across tissues[20–22]. SOX9 is often overexpressed in various solid tumors and its upregulation contributes[23] to the mutability of cancerous cells while Sp1 is a ubiquitously expressed transcription factor with binding sites in the promoter regions of many cell cycle regulatory proteins[24]. In combination, these observations support the role of genome instability in the origin of de novo variation while characterizing a background of de novo TR mutations in the UKB cohort.

The most significant PheWAS finding related *NCOA6*-[GT]$_N$ and *ease of skin tanning*. This TR was enriched for association with dermatological phenotypes and was determined likely causal for ease of skin tanning, hair color, and skin color, with effects independent of all surrounding SNPs and indels. *NCOA6* gene expression has been previously associated with outwardly visible characteristics[25] and variation in this locus is part of predictive algorithms for freckling in individuals of European ancestry[26]. Here, we describe the likely causal effect of an intronic TR that may hold additional predictive properties that explain the poor genetic prediction of intermediate phenotypes due to potential regulatory features encoded by this region. For example,

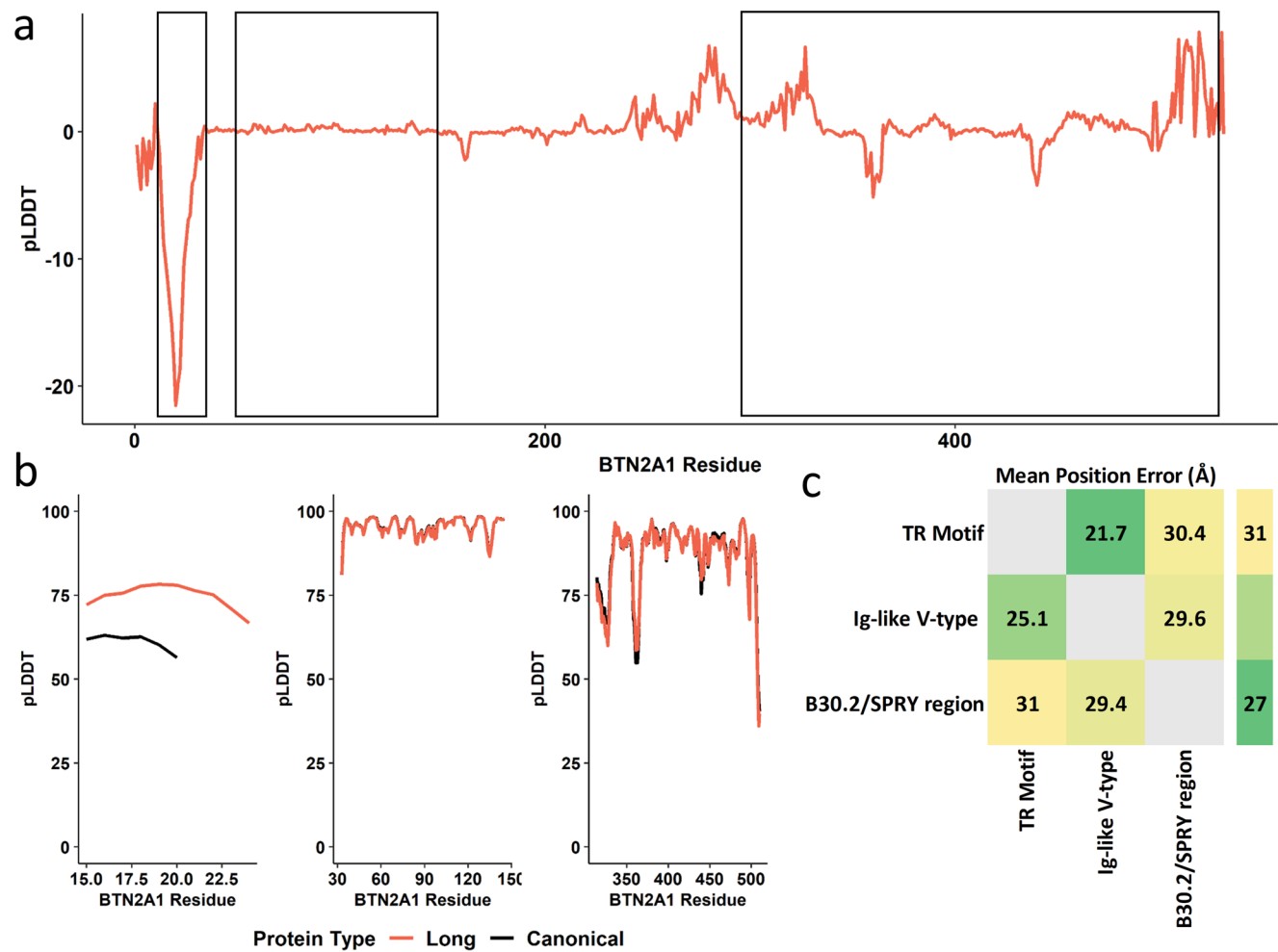

**Fig. 7 | Structural changes in BTN2A1 due to TR variation.** Putative impact of tandem repeat mutations on the structure of BTN2A1. The per-residue folding confidence in canonical BTN2A1 was compared to the long mutation (*BTN2A1*-[CCT]$_{10}$) (**a**) and zoomed-in regional confidence estimates for protein structures are shown in **b**. In **c**, the canonical (bottom left) and mutated (top right) predicted alignment errors are compared between the TR-containing motif, the Ig-like V-type region, and the B30.2/SPRY region. Cells occupied by text indicate a pairwise significant difference in angstrom (Å) between canonical and mutated BTN2A1. Additional details for **c** are provided in Supplementary Data 12. Abbreviations: per-residue predicted local-distance difference test (pLDDT), Butyrophilin Subfamily 2 Member A1 (*BTN2A1*), immunoglobulin-like variable domain (Ig-like V-type).

SNP-based skin color predictions poorly perform with pale and intermediate skin tones but predict darker skin tones relatively well[27].

Fine-mapping TR loci revealed approximately 20% of the TR-phenotype associations detected were likely causal such that nearby SNPs and indels were less likely to explain phenotypic variation than the TR itself. Two TRs had relatively large effects on biological features relevant for human health and disease. *BTN2A1*-[CCT]$_N$ encodes a transmembrane helical domain and was associated with isotropic volume fraction in the thalamus. *FAN1*-[TG]$_N$ encodes an intronic protein domain and was associated with the mean and maximum carotid intima-medial thickness. BTN2A1 is a subtype of butyrophilin, an immunoglobulin family associated with chronic kidney disease, ulcerative colitis, and rheumatoid arthritis[28]. Here, we demonstrate that longer repeat copy number in *BTN2A1*-[CCT]$_N$ associated with greater ISOVF in the superior thalamus, a region of the brain also implicated in chronic kidney disease[29], ulcerative colitis[30], and common psychiatric conditions like depression[31]. These pleiotropic effects link comorbid conditions to clear biological mechanisms (i.e., longer repeat lengths correspond to larger thalamus ISOVF).

*FAN1* encodes a DNA repair nuclease and has been previously implicated in several TR-associated health outcomes[32,33]. The most notable is the polyglutamine expansion underlying Huntington's disease and spinocerebellar ataxia type 1[34]. Though itself a nuclease, FAN1 may modulate TR instability through a nuclease-independent mechanism related to its own genetic variation but other data suggest certain aspects of nuclease activity contribute to repeat instability during development[32,33]. The *FAN1*-[TG]$_N$ intronic TR was associated with a likely causal effect on carotid IMT independent of common SNPs that empirically influence FAN1 activity (e.g., rs3512 and rs35811129)[32,33]. The large effect at this locus corresponds to a 78.7-micrometer difference in mean and 98.6-micrometer difference in maximum carotid IMT. Carotid IMT above 880-micrometers has been ascribed atherosclerosis diagnostic sensitivity and specificity up to 88% and 90%, respectively[35], though these metrics range with demographics of the sample. Taken together, genotyping the *FAN1*-[TG]$_N$ TR may have the potential to increase the diagnostic accuracy of atherosclerosis given carotid IMT data and, more broadly, may inform effects of trans-acting TR-containing genes expressed during development and/or adulthood[32,33].

Motivated by the relationship between protein structure and function, we used structural prediction to investigate the effects of exonic tandem repeat mutations on 3-dimensional attributes of the protein with the ambition of identifying putative functional effects of these loci. The range of alleles observed at *FNBP4*-[GGT]$_N$ encodes a stretch of threonine residues. The TR contraction observed in *FNBP4* associated with structural changes elsewhere in the protein such that

formin-binding sites may be destabilized in the presence of *FNBP4*-[GGT]$_2$ relative to the canonical *FNBP4*-[GGT]$_5$. We hypothesize that this destabilized binding pocket may increase affinity for formin. As formin and FNBP4 interaction is essential for cytoskeletal formation, the association between *FNBP4* and limb size measurements (Supplementary Data 8) is consistent with mutations in this gene conferring rare disorders with prominent limb abnormalities[36].

We also identified a TR-phenotype association that supports a priori biology of neurodegenerative disease. *HTT*-[CAG]$_N$ was mutated in just a single family. However, when subjected to our population-scale PheWAS, we identified a relationship with mean ISOVF in the cerebellar peduncle. Neuronal loss in the cerebellum is routinely implicated in Huntington's pathology[37]. Given the ages of the participants from the family harboring this mutation, the relatively short *HTT*-[CAG] alleles they carry, and the small effect size detected, further targeted assessment of the CAG-repeat is required to determine if this region may be associated with nontraditional clinical features that may hold pre-symptomatic diagnostic value[38].

This study has some limitations to consider. First, TRs are abundant throughout the genome and similar to SNP signals detected by GWAS, non-coding TRs may play regulatory roles through trans-QTL effects that cannot be captured by exome-focused studies of the genome. Furthermore, the complex traits assessed herein are highly polygenic and although several TRs identified in this study have relatively high effects, they likely act in aggregate rather than isolation. Future studies are required to model the polygenic versus monogenic burden of TRs, SNPs, and the combination of TRs and SNPs. Second, the de novo mutation background characterized in this study is likely underestimated due to the relatively small number of trios and the focus on European ancestry participants. Future studies will leverage the full extent of family data and ancestry diversity in the UKB to quantify the diversity of de novo TR mutations and their cross-ancestry heterogeneity. Additionally, future work may leverage other deeply phenotyped population and family data to more robustly quantify the de novo TR landscape. Third, though GangSTR[39] was designed to call TRs from short-read sequencing chemistries, longer TRs and/or complex genic regions of the genome may be better investigated with long-read sequencing chemistries and appropriate alignment strategies to better resolve de novo variation in these regions. Furthermore, GangSTR's error rate is largely attributed to dinucleotide repeats with AC/TG motifs. Though conferring large effects on phenotypic outcomes in this study, the relatively small error rate from GangSTR for this locus motif may bias the accuracy of their effect estimate[40]. Finally, we report TR-trait associations with high fine-mapping probabilities, prioritized further based on ability to confer structural changes suggesting they alter protein confirm causal effects of these TRs on phenotype variation. Further work is required to functionally validate these results in model systems.

This study provides an atlas of the phenotypic spectrum associated with loci harboring de novo TR mutations in the UKB. Independent of the surrounding biallelic genetic variation, the TRs detected here have relatively large effects on phenotypic outcomes and generally localize to regions that translate to amino acid stretches in proteins. Therefore, these associations serve as testable hypotheses regarding the size and copy number effects of TR loci on health and disease with clear dose-dependent implications. These findings will contribute to the mechanisms linking genetic variation, protein structure, and health and disease outcomes.

## Methods
### UK Biobank
The UKB is a large population-based cohort of >500,000 participants. UKB assesses a wide range of factors in including physical health, anthropometric measurements, circulating biomarkers, and socio-demographic characteristics[41]. This research has been conducted in the scope of UKB application reference number 58146. Individual-level data are available to bona fide researchers through approved access. Ancestry in the UKB was assigned using a random forest classifier based on features predictive of ancestry in a combined 1000 Genomes Project plus Human Genome Diversity Panel Ref. 42. This procedure resulted in 174,371 European, 2885 African, 1220 East Asian, 4107 Centra/South Asian, 692 Middle Eastern, 442 Admixed American participants with whole-exome sequencing (WES) data.

### Family trios
To determine relatedness among participants with WES data, we performed identify-by-descent (IBD) analysis per ancestry group. Linkage disequilibrium (LD) independent SNPs with minor allele frequency >5% were clumped in Plink v1.9[43] using 200-kb window size per 100 variants and a pairwise-$r^2$ of 0.2. Pairwise IBD was performed in Plink using the --genome flag. With all pairs of individuals with pi_hat > 0.2, we used age, sex, and IBD metrics to determine parent-offspring relationships. The following cutoffs were applied to IBD metrics: $0 \leq Z0 \leq 0.05$, $0.75 \leq Z1 \leq 1$, and $0 \leq Z2 \leq 0.05$[44].

### De novo variant calling
UKB CRAM files containing WES reads aligned to the hg38 reference genome were converted to binary alignment map (BAM) files using cramtools v3.0 (February 2021)[45]. Genotyping of autosomal tandem repeats from short reads was performed with GangSTR v2.5.0[39] using sorted and indexed BAMs. Each family trio was processed with a separate GangSTR job resulting in per-family variant call files (VCF). Per-family VCFs were subjected to quality control using dumpSTR[46] to remove TRs with (i) read depths <20X, (ii) reads that only span or flank the genotyped region, and (iii) maximum likelihood genotypes outside the 95% confidence interval reported by GangSTR. Prior data support an genotyping error rate of approximately 0.53% after applying these developer-specified parameters.

MonSTR v1.1.0[13] was used to identify de novo TRs per family. MonSTR evaluates the joint likelihood of genotypes in a parent-offspring trio to estimate a posterior probability of mutation at each TR in the offspring. Through simulation, it was previously demonstrated that MonSTR has a low false positive rate (<1%) for mutations of up to 10 repeat copies[13]. MonSTR was run per family with non-default parameters as performed previously including the following flags:--max-num-alleles 100, --gangstr, --min-total-encl 10, --posterior-threshold 0.5, and --default-prior −3[13]. In combination, these flags remove error-prone TRs, indicate to use GangSTR-output likelihoods, apply a constant prior of per-locus mutation rate set to $10^{-3}$, require de novo mutations to be supported by at least three enclosing reads, require a minimum of 10 enclosing reads per sample in the family, and label calls with posterior probability ≥0.5 as mutations.

### Gene annotation
TR mutations were positionally mapped to genes using the hg38 reference genome. To identify LD-independent genomic regions, we retained only one TR-containing gene in a 200-kb window. When TR-containing genes were closer than 200-kb[15], we retained the gene with the larger number of de novo mutations observed. Hypergeometric tests were used to determine gene set enrichments of gene ontologies from the Molecular Signatures Database v7.0 using Functional Annotation and Annotation of Genome-Wide Association Studies (FUMA v1.3.7). The number of gene-sets per category were 50 for Hallmark gene sets, 299 for positional gene sets, 6366 for curated gene sets, 3726 for regulatory gene sets, 85 for computational gene sets, 15,473 for ontology gene sets, 189 for oncogenic gene sets, 5219 for immunologic gene sets, and 700 for cell-type signature gene sets. Multiple testing correction was performed per gene set category using a false discovery rate of 5%[47]. The same gene-sets were then tested with a second method, ShinyGo v0.65[48], which was used as a complementary

method because it reports the magnitude of the fold change associated with each gene set. ShinyGo reports *P*-values that have been adjusted using a false discovery rate of 5% considering all gene sets tested in the platform.

### Expression associated TRs
Gene expression associated TRs (eSTR) were identified using data from ref. 15. Briefly, average TR repeat length, called from whole genome sequences using HipSTR[6], was associated with gene expression in 17 tissues from the Genotype-Tissue Expression Project ($N = 652$ unrelated individuals; 86% European ancestry). All TR-gene expression relationships were adjusted for age, sex, and the top 10 within-ancestry principal components. Multiple testing correction was performed using the false discovery rate method (5%) considering 172 genes x 17 tissues = 2924 tests.

### Phenome-wide association (PheWAS)
PheWAS was performed in up to 148,607 unrelated UKB participants of European ancestry based on the random forest classifier applied in the Pan-ancestry UK Biobank project[42]. Linear and logistic regression models were applied to continuous and binary outcomes, respectively, with the requirement for at least 1500 cases for binary traits (-1% of the total sample size). TR loci with de novo mutations were called from WES data using GangSTR, as described above in De novo *variant calling*. PheWAS was performed for all loci harboring de novo TRs requiring a mean read depth >20X and missingness <5% across all unrelated European ancestry participants. After site filtering, the mean read depth per individual was 78.5X ± 16.3. After application of read alignment and variant calling settings in GangSTR, we expect an UKB TR genotyping error rate of approximately 0.53%.

After trait, individual, and TR quality control steps, PheWAS was performed for 416 TRs across 1844 phenotypes. To associate multi-allelic TRs with traits of interest, we converted TR genotypes to a TR length sum which captures locus-level burden. The length sum per TR was calculated by adding the number of repeat units in a genotype. Prior to regression, the TR length sum per locus was normalized/residualized using age, sex, sex × age, age$^2$, sex × age$^2$, and the top 10 within-ancestry principal components as covariates. The generalized linear model function in R was used to regress normalized/residualized TR length burden on each phenotype. Multiple testing correction was applied using the false discovery method (false discovery rate (FDR) <5%) to accommodate the correlation between traits and between TRs (767,104 total tests performed).

### Fine-mapping TR-SNP signals
To determine if TR-phenotype association signals represented likely causal relationships between TR variation, rather than tagging nearby SNP signals, we fine-mapped each TR-containing locus[8]. We computed the linear regression between TR length sum and all SNPs and insertion/deletion (indel) polymorphisms within 500-kb of the TR start site. Bayesian fine-mapping was performed with FINEMAP v1.3.1[49] with the optional flags: --corr-config 0.999 --sss --n-causal-snps 5. These settings estimate the likelihood of causality for the TR accounting for LD with other genetic variants in the region. SNP and indel effect sizes per trait were estimated by linear or logistic regression in the same sample from which TR effect sizes were estimated. Each SNP/indel effect size estimate included age, sex, sex × age, age$^2$, sex × age$^2$, and the top 10 within-ancestry principal components as covariates.

### Protein domain annotation and 3-dimensional structure
For each TR with a high fine-mapping posterior probability, we characterized the protein domain containing this motif. When a TR was found within an exon of the containing protein, we annotated the amino acid residue string sequence to protein domains using InterPro[50].

Next, we evaluated how the presence of mutated forms of the TR may influence protein function relying on the notion that protein function is intimately linked to protein structure. Using AlphaFold v2.2.0[16] with default parameters, we predicted the structure of a subset of proteins containing exonic TRs with fine-mapping probabilities >0.95. Each mutated form of the protein was compared to the canonical sequence accessible in UniProt[51]. We predicted various forms of each protein by using the shortest and longest TR allele lengths with frequencies >1% in the European ancestry population from UKB. We compared the per-residue pLDDT scores of the mutated proteins back to the canonical sequence to identify changes in local protein folding confidence. Among the regions with altered folding confidence, we compared the PAE of a given region between the different versions of the protein. Briefly, pLDDT is a per-residue confidence metric that evaluates local distance differences of all atoms in a model, including stereochemical plausibility[16]. As a measure of local model quality, pLDDT less than 50 is considered a strong predictor of structural disorder[16]. Predicted alignment error (PAE) is a metric evaluating relative orientation of two residues in 3-dimensional space. If the PAE is generally low for a residue pair, relative positions and orientation of those two residues are well-defined[16].

### Reporting summary
Further information on research design is available in the Nature Portfolio Reporting Summary linked to this article.

## Data availability
All genotype-phenotype association data generated during this study are included in this published article and its supplementary information files. The individual-level genotype data are available under restricted access to preserve participant privacy, access can be obtained by bona fide researchers through the UK Biobank Data Analysis Platform. The data for this project were accessed through approved protocol 58146.

## Code availability
No custom code was developed for this project. We used previously developed pipelines for the following analytic procedures:
- converting .cram to .bam (cramtools v3.0, https://stab.st-andrews.ac.uk/wiki/index.php/Cramtools)
- calling tandem repeat genotypes from DNA sequence reads using GangSTR v2.5.0 and MonSTR v1.1.0 (https://gymreklab.com/).
- kinship assignment (PLINK v1.9, https://www.cog-genomics.org/plink/)
- finemapping with FINEMAP v1.3.1 (http://www.christianbenner.com)
- assessing protein structural changes with AlphaFold v2.2.0 (https://alphafold.ebi.ac.uk)
- gene set enrichment with ShinyGO v0.65 (http://bioinformatics.sdstate.edu/go/#tab-1377-10) and FUMA v1.3.7 (https://fuma.ctglab.nl)
- generalized linear models in R (v 4.0.3) for association of tandem repeats with phenotypic data
- annotation of protein domains with InterPro 90.0 (https://www.ebi.ac.uk/interpro/release_notes/)
- annotate canonical protein sequences with UniProt (release 2022_04, https://www.uniprot.org)

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

## Acknowledgements

This research was conducted using the UK Biobank Resource (application reference no. 58146). The authors thank the research participants and employees of the UK Biobank for making this work possible. This study was supported, in part, by the National Institutes of Health (R21 DC018098 to RP, R33 DA047527 to RP, and F32 MH122058 to FRW).

## Author contributions

F.R.W. conceptualized the study design, carried out the analysis, and drafted the manuscript. G.A.P. provided feedback on statistical analyses and critical review of the manuscript and all data. R.P. conceptualized the study design, provided feedback on statistical analyses, and provided critical review of the manuscript and all data.

## Competing interests

The authors declare no competing interests.
