## [Peer Review File · Nature Communications]

Phenome-wide association study of loci harboring de novo tandem repeat mutations in UK Biobank exomesREVIEWER COMMENTS

Reviewer #1 (Remarks to the Author):

This manuscript describes a phenome-wide association study (PWAS) focusing on tandem repeats (TRs) and in particular de novo mutations. It uses the rich resource of whole-exome sequencing datasets from the UK Biobank. The work is novel and important. The manuscript is generally well written. I have suggestions below for improvements to the manuscript.

1. The final analysis was focused on 39 trios of European descent, with 34 of these having at least one de novo TR mutation (with an average of over 30 de novo TR mutations per family). Whilst I realise that the real power and novelty of this study is the deep phenomic data (associated with the UK Biobank; 1,844 human traits) which facilitates PWAS, do the authors feel this is a sufficient number of trios to ensure robustness and replicability?

2. Related to the above point, are the authors able to analyse independent collections of whole exome (including data derived from whole genome sequences) sequences associated with phenomic data. Whilst I realize other collections would not have the same phenomic data as the UK Biobank, any additional associations with de novo TR mutations in trios would be informative.

3. And as a follow-up point, are the authors able to perform a PheWAS not just on the TR mutations within the trios, but across all 174,371 European exomes (with associated phenomic data) in the UK Biobank? This is likely to identify many more TRs which, although not necessarily as mutable in the germline (between generations), could contribute to the 'missing heritability' of such phenomes/traits. An analogy is that the authors cite Mitra et al. (Nature, 2021) who used trios and DNA sequence analyses to associated TR mutations with autism, however there was another recent study (Nature 2020; 586(7827):80-86. PMID: 32717741) which compared cases and controls and found a different group of TRs associated with autism (as discussed: Nature 2021; 589(7841):200-202. PMID: 33442037).

4. The authors report that 'the maximum mutation was 6 repeat units at HTT-[CAG]N'. Of course the CAG/glutamine repeat expansion in the huntingtin (HTT) gene/protein is one of the best-studied of all TRs, due to its expansion in Huntington's disease. Can the authors elaborate further on the phenotypic associations with polymorphism in the CAG repeat within the HTT gene, in the context of the HTT and Huntington's disease literature? What was the length of the expanded CAG repeat in the HTT gene of the offspring, what was their age, and could their phenotypic associations be related to presymptomatic Huntington's disease?

5. The authors state that 'The poly-leucine stretch encoded by this TR makes up a portion of the NOTCH4 extracellular domain and therefore may contribute to abnormal astrocyte differentiation, angiogenesis, and coronary vessel development'. Can the authors use a recently developed protein structure/function prediction tools, such as AlphaFold, to model the structure of NOTCH4 with the different poly-leucine lengths associated with the TR mutations in their trios? Does changing the poly-leucine tract length change structure (and function) of NOTCH 4 in any way? Can they do this for other protein-encoding TRs discussed in the present manuscript?

6. In the Discussion, whenever the authors discuss a particular TR mutation and associated gene, they should be specific as to where the TR is located in the gene, whether it is predicted to be coding amino acids within an open reading frame (as the authors note for the poly-leucine tract in NOTCH4), and what effect such a TR mutation is likely to have on the respective gene/RNA/protein. For example for BTN2A1-[CCT]N, the authors do not provide any detailed discussion of what molecular (i.e., epigenetic, gene expression, RNA structure/function and/or protein structure/function) effects a mutation in this CCT repeat might have. Similarly, there is a lack of such a discussion for the FAN1-[TG]N TR mutation.

7. One interesting thing about FAN1 is its potential regulation of somatic TR repeat instability, including during development (Cell Rep. 2021; 37(10):110078. PMID: 34879276). Thus the FAN1 TR association in the present manuscript could relate to trans-acting effects on other TR-containing genes, associated with human development and/or adult function. This is perhaps worthy of a sentence or two of discussion.

8. There is a question as to whether short-read sequencing of exomes (or genomes) is ideally for such TR-focused studies. There is evidence long-repeat sequencing can pick up additional mutations (e.g. Am. J. Hum. Genet. 2022; S0002-9297(22)00065-9. PMID: 35290762). Perhaps this could be mentioned as an additional limitation.

9. A role for TR mutations and polymorphisms in finding the 'missing heritability' (resulting from SNP-based GWAS) for human traits and diseases was first proposed a dozen years ago (Trends Genet. 2010; 26(2):59-65. PMID: 20036436). The authors should acknowledge this prior article, as their manuscript provides important evidence for TRs explaining at least some aspects of this missing heritability. This is one of the major implications of their study.

10. The statement in the Introduction needs to be corrected:

'Huntington's disease, which is characterized by over 100 copies of a CAG motif in HTT'. The threshold of CAG repeats is in the high 30's, with 40 or more being fully penetrant for Huntington's disease (and a range of around 35-39 repeats showing incomplete penetrance).

11. Minor corrections:

Abstract, correct to 'to assess the phenome-wide impact of de novo TRs'

P3, line 41, add full stop after 'several biomarkers of human health'.

12. In the supplementary Table S1, the table itself is split across four pages of the pdf, and the title is split across the first two pages. Whilst I was able to easily work this out, correct formatting of the file should be ensured before publication, to make it easy and intuitive for the reader.

Reviewer #2 (Remarks to the Author):

The study by Wendt et al analyses exome sequencing data from the UK Biobank to perform a phenome-wide association analysis of tandem repeats (TR). The authors first identify de novo TR mutations in 39 trios, and then perform a number of TR association analyses for these de novo TR loci in the wider UKB dataset. These association analyses include gene set enrichment tests, using gene ontologies from the Molecular Signatures Database, and an identification of TR-eqtls using GETX data. The authors then perform a phenome-wide association analysis of the de novo TR loci for 1,844 traits in the wider UKB dataset, and identify 426 TR-phenotype associations, the most significant of which is a TR associated with 'ease of skin tanning'. Finally, the authors use finemapping to refine 123 TR-phenotype associations to a set of 41 likely casual TRs.

A strength of the current study is that it is a population-based investigation of TR, which are underrepresented when compared with the wealth of population-based studies of SNPs or rare coding variants. The study also reports a number of novel TR-phenotype associations, which if true provide important biological insights into phenotypes reported. However, the study has several weaknesses which are outlined below.

1) Firstly, I find the study design a bit confusing. Can the authors more clearly outline the rationale for performing a TR phewas only for those TR de novo loci identified in a very small trio sample (n = 39)? I'm aware of previous studies that have tested de novo CNVs for association in independent case-control data, but I think this approach is more applicable for identifying ultra-rare variants that are associated with phenotypes under strong selection (PMID: 18668039). In the current work, the authors do not focus on rare TRs in the wider phewas analysis, and it is therefore unclear to me why they chose this study design. Why did the authors not just perform a TR-phewas of all TRs in the wider UKB analysis? To help the reader understand why they chose this approach, I would ask for the authors to clearly outline the rationale for choosing this study design in the introduction.

2) Aligned to the above comment, given the phewas is based on de novo TRs identified in 39 trios, are they enriching for TR-phenotype associations for traits observed among these trios?

3) A large number of tests are performed in both the gene-set analysis, but it is hard to identify from the paper exactly how many independent tests are performed. For example, how many genes sets were in each MSigDB category? Were all gene-sets among all 17 MSigDB categories corrected for? For both the gene set analysis and other analyses reported in the manuscript, the exact number of independent tests performed should be clearly outlined in the methods.

4) De novo mutations are known to be enriched for false positives. This is true for de novo SNVs, indels, CNVs and TRs. In the current manuscript, I have no idea what the likely false positive rate is for the de novo TRs that form the basis of this work. I appreciate the limitations with analysing UKBB data for experimentally validating the TRs, but are there any other means by which the authors could validate a subset of their mutations? For example, could they use any whole genome sequencing data from the same participants? This would be particularly beneficial for providing more confidence that top TR associations reported in the abstract and paper and indeed based on true TRs, and not false positive mutations. Moreover, a lack of experimental validation data should be included as a limitation in the discussion.

5) I could not find a detailed description of the quality control used for excluding low quality samples, low quality TR loci, and low-quality TR genotypes. Line 298 states “After trait, individual, and TR quality control steps”, but I could not find these methods or data. Some QC is reported for the de novo analysis, but I could not find detailed information for the wider UKB analysis. Apologies if I have simply missed this, but if not, this is crucial information that should be included in the manuscript.

Minor point

1) Line 36 makes it sound like Huntington’s disease is characterised by having over 100 copies of the CAG motif, but having 40 copies is strongly associated with HD.

Reviewer #3 (Remarks to the Author):

This interesting article by Wendt, Pathak, and Polimanti examine de novo tandem repeat mutations within the Whole Exome Sequencing Data of the UK Biobank. From 40 trios, the authors identify 1031 de novo tandem repeat mutations within the probands of 34 European-descent trios. These repeats were annotated, and a phenome-wide association study was conducted across 1844 UK Biobank phenotypes. The general premise of this work is interesting and significant, as repeat polymorphisms are an often overlooked component of genetic variation that likely influences a variety of human traits. I have a number of concerns, however, that diminish my enthusiasm for this work as presented.

1. More information is needed on the quality control, validation, and accuracy of GangSTR calls from whole exome data. Some reports suggest that the performance of this method may only be satisfactory for repeats of 3-6 bp in length (PMID: 32665844). How are missing genotypes called? Because this study has the added complication of detecting de novo events, understanding the quality of the genotype calls is even more important, and it is difficult to assess the overall significance of the work without higher confidence in the repeat genotyping.

2. The methods for the PheWAS analysis are difficult to follow. The authors mention 148,607 unrelated UKBB participants, and linear/logistic regression on the outcomes. Was this analysis conducted on the 30 probands with de novo STR calls relative to all other participants? How did the authors account for what must have been extremely small cell counts? Did the authors include the sequenced parents of the proband in the analysis, or account for relatedness? The authors state that the length-sum was normalized for covariates - why was the phenotype not normalized for these covariates?

3. In examining the supplemental tables, the authors appear to have conducted PheWAS association tests across the full sample set, which raises several other questions - if there are thousands of people with Notch4 repeats, why are the authors calling this a de novo mutation? Are the authors considering the NUMBER of repeats to be a "de novo" allele? If so, won't all participants have "de novo" repeats SOMEWHERE in the genome?

In sum, this work has tremendous potential and is straightforward in its goals, but it is difficult to follow and have confidence in the results as presented.

10 August 2022

Ref: NCOMMS-22-04483 revised manuscript submission

We would like to thank the Editor and Reviewers for their careful and thorough reading of our manuscript and for their supportive comments, which helped to improve its quality. We have revised the text considering their useful suggestions and comments.

We hope our revision has improved the paper to a level of their satisfaction.

Please find below the response to all the Reviewers' comments.

Reviewer #1 (Remarks to the Author):

This manuscript describes a phenome-wide association study (PheWAS) focusing on tandem repeats (TRs) and in particular *de novo* mutations. It uses the rich resource of whole-exome sequencing datasets from the UK Biobank. The work is novel and important. The manuscript is generally well written. I have suggestions below for improvements to the manuscript.

Reviewer Comment 1: The final analysis was focused on 39 trios of European descent, with 34 of these having at least one *de novo* TR mutation (with an average of over 30 *de novo* TR mutations per family). Whilst I realize that the real power and novelty of this study is the deep phenomic data (associated with the UK Biobank; 1,844 human traits) which facilitates PWAS, do the authors feel this is a sufficient number of trios to ensure robustness and replicability?

Author Response: We think this comment from the Reviewer may stem from the unclear description of our study design. We now include an overview figure (Figure 1) in our resubmission to clarify that our analysis used family trios only to identify *de novo* variants. In line with Reviewer #1's comment, we recognized the low power of this small cohort for PheWAS (page 5 paragraph 3). Accordingly, we used the loci containing *de novo* variants from these trios to perform PheWAS in the remaining UKB European ancestry sample (148,607 unrelated individuals). Certainly other *de novo* TRs exist elsewhere in the genome in different trios of different ancestries but the association between TR and phenotype should be robust as this relies on extremely large population-based association tests. We now also add text to our discussion section clarifying that while the associations are robust, we do not capture the full scope of loci that contain *de novo* TRs and require more family trios to ascertain this information (page 11, paragraph 3).

Reviewer Comment 2: Related to the above point, are the authors able to analyse independent collections of whole exome (including data derived from whole genome sequences) sequences associated with phenomic data. Whilst I realize other collections would not have the same phenomic data as the UK Biobank, any additional associations with *de novo* TR mutations in trios would be informative.

Author Response: We appreciate this suggestion. We searched dbGaP for all cohorts with “parent-offspring trios” listed as their study type to match the study design from the UK Biobank. Of the eight studies with >50 participants, only five have general research use consent options that would permit access to address the Reviewer’s comment. Of these, only three studies were performed using whole exome or whole genome sequencing for a maximum sample size (before quality control and ancestry assessment) of 1,544 trios. Unfortunately, these cohorts have very narrowly ascertained phenotype data. Studies phs001436 and phs001228 target family cancer history, which may bias *de novo* variant calling. The dataset phs000687 is a study of schizophrenia families with no other phenotype data collected. We appreciate the desire for larger external studies of the impact of *de novo* TRs but, at present, it appears these data are not readily available for deeply phenotyped cohorts like the UK Biobank. We clarified the lack of additional whole-exome or whole-genome sequencing data to extend the findings we obtained from UKB cohort (page 11 paragraph 3).

Reviewer Comment 3: And as a follow-up point, are the authors able to perform a PheWAS not just on the TR mutations within the trios, but across all 174,371 European exomes (with associated phenomic data) in the UK Biobank? This is likely to identify many more TRs which, although not necessarily as mutable in the germline (between generations), could contribute to the ‘missing heritability’ of such phenomes/traits. An analogy is that the authors cite Mitra et al. (Nature, 2021) who used trios and DNA sequence analyses to associated TR mutations with autism, however there was another recent study (Nature 2020; 586(7827):80-86. PMID: 32717741) which compared cases and controls and found a different group of TRs associated with autism (as discussed: Nature 2021; 589(7841):200-202. PMID: 33442037).

Author Response: We appreciate this comment and wish to clarify that we did not perform PheWAS in the trios. We used these trios to find loci with *de novo* TR mutations and then performed a PheWAS of those loci in the remaining 148,607 unrelated European ancestry participants with whole exome sequencing data available (Figure 1). Without family data for the remaining samples, we cannot confidently identify an allele as *de novo* in the remaining European ancestry participants. Furthermore, while we absolutely agree that a PheWAS of all TRs in the exome is of great interest to the field, this suggested analysis distracts from the *de novo* variant emphasis of our study. Additionally, it may create confusion among the readers, because it would be analogous to presenting 1,844 genome-wide association studies in a single article. Indeed, many leaders in this area are studying the PheWAS of a subset of TRs (Mukamel, et al. PMID: 34554798) or a genome-wide assessment of TRs in a subset of phenotypes (Margoliash, et al. bioRxiv 2022.08.01.502370). With respect to Mukamel, et al., we articulate findings for more loci and more phenotypes. With respect to the recent work of Margoliash, et al., many of their hematology findings are already reported in our Supplementary Material as our *de novo* TRs are largely enriched for hematological signals. Both approaches permit the presentation of more clear stories which is paramount for the novelty and impact of this work. For these reasons, we have not included this analysis in our resubmission as we believe it is far beyond the scope of the original submission and would be difficult to present as a single, cohesive, and concise collection of findings.

Reviewer Comment 4: The authors report that ‘the maximum mutation was 6 repeat units at HTT-[CAG]_N’. Of course the CAG/glutamine repeat expansion in the huntingtin (HTT) gene/protein is one of the best-studied of all TRs, due to its expansion in Huntington’s disease. Can the authors elaborate further on the phenotypic associations with polymorphism in the CAG repeat within the HTT gene, in the context of the HTT and Huntington’s disease literature? What was the length of the expanded CAG repeat in the HTT gene of the offspring, what was their age, and could their phenotypic associations be related to presymptomatic Huntington’s disease?

Author Response: This is a great suggestion. We have added language to highlight this result:

- We corrected a typographical error in the Results section *Family trios and de novo mutations* (page 4 paragraph 2). We incorrectly listed the *HTT*-[CAG]_N *de novo* repeat as an expansion when it was a contraction and the *DST*-[CA]_N *de novo* repeat as a contraction when it was an expansion.
- We added a Results section titled *Other notable loci from the TR literature* where we highlight the presence of just one family harboring an *HTT*-[CAG]_N *de novo* contraction (page 7 paragraph 1). None of the family members have a motif that suggests a Huntington’s phenotype (allele sizes range from 9-17). *HTT*-[CAG]_N had one significant association in our PheWAS with mean ISOVF in the right inferior cerebellar peduncle.
- We added a Discussion paragraph to place these findings into a broader context. Given the small sample size (one family), small effect size, and short CAG-allele spread in the family of interest, we use cautious language about the potential detection of pre-symptomatic Huntington’s disease (page 11 paragraph 1).

Reviewer Comment 5: The authors state that ‘The polyleucine stretch encoded by this TR makes up a portion of the NOTCH4 extracellular domain and therefore may contribute to abnormal astrocyte differentiation, angiogenesis, and coronary vessel development’. Can the authors use a recently develop protein structure/function prediction tools, such as AlphaFold, to model the structure of NOTCH4 with the different polyleucine lengths associated with the TR mutations in their trios? Does changing the polyleucine tract length change structure (and function) of NOTCH 4 in any way? Can they do this for other protein-encoding TRs discussed in the present manuscript?

Author Response: We really appreciate the Reviewer’s suggestion regarding predicted protein folding and have added these analyses to our article. For each gene in Table S8 that contains an exonic TR with high fine-mapping probability relative to a given trait, indicating a likely causal relationship, we tested various protein models in AlphaFold. We provide details of this analysis in the text. Our most interesting observation relates to *FNBP4* where the TR repeat contraction alters the structure of the FNBP4 binding motif. Please refer to these results (page 7-8), discussion (page 10 paragraph 3), Figures 6 and 7, and the Supplementary Material documentation.

Reviewer Comment 6: In the Discussion, whenever the authors discuss a particular TR mutation and associated gene, they should be specific as to where the TR is located in the gene, whether it is predicted to be coding amino acids within an open reading frame (as the authors note for the polyleucine tract in NOTCH4), and what effect such a TR mutation is likely to have on the

respective gene/RNA/protein. For example for BTN2A1-[CCT]N, the authors do not provide any detailed discussion of what molecular (i.e., epigenetic, gene expression, RNA structure/function and/or protein structure/function) effects a mutation in this CCT repeat might have. Similarly, there is a lack of such a discussion for the FAN1-[TG]N TR mutation.

Author Response: We appreciate this suggestion and have used InterPro (Blum, et al. 2021; PMID:33156333) to assign each TR with a high fine-mapping probability (Table S8) to a protein domain. Table S9 now summarizes these annotations and we incorporate this information at each point of discussion for a given protein (page 9 paragraphs 3 and 4 and page 10 paragraph 2).

Reviewer Comment 7: One interesting thing about FAN1 is its potential regulation of somatic TR repeat instability, including during development (Cell Rep. 2021; 37(10):110078. PMID: 34879276). Thus the FAN1 TR association in the present manuscript could relate to trans-acting effects on other TR-containing genes, associated with human development and/or adult function. This is perhaps worthy of a sentence or two of discussion.

Author Response: We've incorporated this information into our discussion of FAN1 and thank the Reviewer for their insight on this topic (page 10 paragraph 2).

Reviewer Comment 8: There is a question as to whether short-read sequencing of exomes (or genomes) is ideally for such TR-focused studies. There is evidence long-repeat sequencing can pick up additional mutations (e.g. Am. J. Hum. Genet. 2022; S0002-9297(22)00065-9. PMID: 35290762). Perhaps this could be mentioned as an additional limitation.

Author Response: This is a great point as well. We intentionally chose the GangSTR pipeline which uses a maximum likelihood estimate of TR length. Though designed for shorter read chemistries, we appreciate the potential of misaligned or inappropriately called TRs that could be better resolved from long-read sequencing chemistries and have included such language in our limitations section (page 11 paragraph 3).

Reviewer Comment 9: A role for TR mutations and polymorphisms in finding the 'missing heritability' (resulting from SNP-based GWAS) for human traits and diseases was first proposed a dozen years ago (Trends Genet. 2010; 26(2):59-65. PMID: 20036436). The authors should acknowledge this prior article, as their manuscript provides important evidence for TRs explaining at least some aspects of this missing heritability. This is one of the major implications of their study.

Author Response: We appreciate this suggestion and have added this reference and language regarding the discovery of missing heritability not captured by GWAS (page 3 paragraph 1 & page 8 paragraph 2).

Reviewer Comment 10: The statement in the Introduction needs to be corrected: 'Huntington's disease, which is characterized by over 100 copies of a CAG motif in HTT'. The threshold of CAG repeats is in the high 30's, with 40 or more being fully penetrant for Huntington's disease (and a range of around 35-39 repeats showing incomplete penetrance).

Author Response: This statement has been corrected to reflect a typical cutoff of 40 CAG repeats being observed in Huntington's disease cases.

Reviewer Comment 11: Minor corrections:

Abstract, correct to 'to assess the phenome-wide impact of de novo TRs' P3, line 41, add full stop after 'several biomarkers of human health'.

Author Response: We have corrected these typographical errors.

Reviewer Comment 12: In the supplementary Table S1, the table itself is split across four pages of the pdf, and the title is split across the first two pages. Whilst I was able to easily work this out, correct formatting of the file should be ensured before publication, to make it easy and intuitive for the reader.

Author Response: We apologize for this inconvenience. The table is submitted as an excel file and the manuscript submission system converts these to .pdf. We will request that the editorial team circulate the excel version to all Reviewers.

Reviewer #2 (Remarks to the Author):

The study by Wendt et al analyses exome sequencing data from the UK Biobank to perform a phenome-wide association analysis of tandem repeats (TR). The authors first identify de novo TR mutations in 39 trios, and then perform a number of TR association analyses for these de novo TR loci in the wider UKB dataset. These association analyses include gene set enrichment tests, using gene ontologies from the Molecular Signatures Database, and an identification of TR-eqtls using GETX data. The authors then perform a phenome-wide association analysis of the de novo TR loci for 1,844 traits in the wider UKB dataset, and identify 426 TR-phenotype associations, the most significant of which is a TR associated with 'ease of skin tanning'. Finally, the authors use finemapping to refine 123 TR-phenotype associations to a set of 41 likely casual TRs.

A strength of the current study is that it is a population-based investigation of TR, which are underrepresented when compared with the wealth of population-based studies of SNPs or rare coding variants. The study also reports a number of novel TR-phenotype associations, which if true provide important biological insights into phenotypes reported. However, the study has several weaknesses which are outlined below.

Reviewer Comment 1: Firstly, I find the study design a bit confusing. Can the authors more clearly outline the rationale for performing a TR phewas only for those TR de novo loci identified in a very small trio sample ($n = 39$)? I'm aware of previous studies that have tested de novo CNVs for association in independent case-control data, but I think this approach is more applicable for identifying ultra-rare variants that are associated with phenotypes under strong selection (PMID: 18668039). In the current work, the authors do not focus on rare TRs in the wider phewas analysis, and it is therefore unclear to me why they chose this study design. Why did the authors not just perform a TR-phewas of all TRs in the wider UKB analysis? To help the reader understand why they chose this approach, I would ask for the authors to clearly outline the rationale for choosing this study design in the introduction.

Author Response: We appreciate this point also brought up by Reviewer 1. We have included an analysis overview figure to clearly show that only detection of *de novo* variants was performed in a set of UKB trios. Appreciating the lack of power to perform a PheWAS in the trios themselves, we took all loci in which a *de novo* TR was observed and assessed what likely impact they have on the phenotypic spectrum, because they can be hotspots for *de novo* mutational events among TR elements (Acuna-Hidalgo, et al. 2016, PMID: 27894357). A PheWAS of all TRs in the exome would distract from the *de novo* variant emphasis of our article. Additionally, it may create confusion in the readers, because it would be analogous to presenting 1,844 genome-wide association studies in a single article. Indeed, many leaders in this area are studying the PheWAS of a subset of TRs (Mukamel, et al. PMID: 34554798) or a genome-wide assessment of TRs in a subset of phenotypes (Margoliash, et al. bioRxiv 2022.08.01.502370). With respect to Mukamel, et al., we articulate findings for more loci and more phenotypes. With respect to the recent work of Margoliash, et al, many of their hematology findings are already reported in our Supplementary Material as our *de novo* TRs are largely enriched for hematological signals. Both approaches permit the presentation of more clear stories which is paramount for the novelty and impact of this work. For these reasons, we have not included this analysis in our resubmission as we believe it is far beyond the scope of the original submission and would be difficult to present as a single, cohesive, and concise collection of findings.

Reviewer Comment 2: Aligned to the above comment, given the phewas is based on *de novo* TRs identified in 39 trios, are they enriching for TR-phenotype associations for traits observed among these trios?

Author Response: The limited number of trios available does not permit us to perform a statistically powerful enrichment analysis (Table S9). Nevertheless, one result was recapitulated in the UKB probands (*USP30*-[CGG]_N versus platelet count, $\beta = -12.62$, $se = 4.99$, $P = 0.017$). This effect was significantly greater than the one observed in the general population ($\beta = -1.39$, $se = 0.159$, $P = 6.17 \times 10^{-19}$; $P_{diff} = 0.025$). These findings are reported on page 6 paragraph 3).

Reviewer Comment 3: A large number of tests are performed in both the gene-set analysis, but it is hard to identify from the paper exactly how many independent tests are performed. For example, how many genes sets were in each MSigDB category? Were all gene-sets among all 17 MSigDB categories corrected for? For both the gene set analysis and other analyses reported in the manuscript, the exact number of independent tests performed should be clearly outlined in the methods.

Author Response: We thank the Reviewer for catching this oversight from our original submission and now add text to clarify the number of tests performed at each stage of analysis.

Reviewer Comment 4: *De novo* mutations are known to be enriched for false positives. This is true for *de novo* SNVs, indels, CNVs and TRs. In the current manuscript, I have no idea what the likely false positive rate is for the *de novo* TRs that form the basis of this work. I appreciate the limitations with analysing UKBB data for experimentally validating the TRs, but are there any other means by which the authors could validate a subset of their mutations? For example, could they use any whole genome sequencing data from the same participants? This would be

particularly beneficial for providing more confidence that top TR associations reported in the abstract and paper and indeed based on true TRs, and not false positive mutations. Moreover, a lack of experimental validation data should be included as a limitation in the discussion.

Author Response: We thank the Reviewer for raising this point. The *de novo* variant calling pipeline applied here has a false positive rate <1% for TR mutations of up to ten copies of the repeat motif (Mitra, et al. 2021; PMID:33442040). We now include this information in the manuscript on page 13 paragraph 2. We also include in the limitations section that experimental validation is required to gain further confidence in these hits; however, the addition of AlphaFold results as requested by Reviewer 1 adds greater confidence to several exonic TR associations (page 11 paragraph 2 and page 12 paragraph 1).

Reviewer Comment 4: I could not find a detailed description of the quality control used for excluding low quality samples, low quality TR loci, and low-quality TR genotypes. Line 298 states “After trait, individual, and TR quality control steps”, but I could not find these methods or data. Some QC is reported for the *de novo* analysis, but I could not find detailed information for the wider UKB analysis. Apologies if I have simply missed this, but if not, this is crucial information that should be included in the manuscript.

Author Response: We include the quality control steps in the Methods section. All TRs analyzed in the population-based PheWAS had read depths greater than 20X and had genotype call rates greater than 95% among European ancestry participants in the UKB. We now also report the mean read depth per sample at the 427 loci subjected to PheWAS: $78.5X \pm 16.3$ (page 13 paragraph 2 and 3). By this metric, all participants had well-sequenced TRs used for association testing.

Reviewer Comment 5: Minor point

Line 36 makes it sound like Huntington’s disease is characterised by having over 100 copies of the CAG motif, but having 40 copies is strongly associated with HD.

Author Response: We appreciate the Reviewer for catching this typographical error and have corrected the text.

Reviewer #3 (Remarks to the Author):

This interesting article by Wendt, Pathak, and Polimanti examine *de novo* tandem repeat mutations within the Whole Exome Sequencing Data of the UK Biobank. From 40 trios, the authors identify 1031 *de novo* tandem repeat mutations within the probands of 34 European-descent trios. These repeats were annotated, and a phenome-wide association study was conducted across 1844 UK Biobank phenotypes. The general premise of this work is interesting and significant, as repeat polymorphisms are an often overlooked component of genetic variation that likely influences a variety of human traits. I have a number of concerns, however, that diminish my enthusiasm for this work as presented.

Reviewer Comment 1: More information is needed on the quality control, validation, and accuracy of GangSTR calls from whole exome data. Some reports suggest that the performance

of this method may only be satisfactory for repeats of 3-6 bp in length (PMID: 32665844). How are missing genotypes called? Because this study has the added complication of detecting *de novo* events, understanding the quality of the genotype calls is even more important, and it is difficult to assess the overall significance of the work without higher confidence in the repeat genotyping.

Author Response: This is a great point. The article referenced by the Reviewer indeed shows GangSTR as a more error-prone approach than other TR genotyping systems (up to 3.3%); however, the authors of this article also note that “once the recommended filters were applied, we found that the performance of GangSTR improved by 2.76% to 0.53%...” and had the most conservative genotype calling rate. In other words, when applied using the developer-recommended procedure, GangSTR reports fewer genotype calls than other genotype methods but these calls have a low error rate. With respect to *de novo* variant quality, we now describe the false positive rate for the GangSTR-MonSTR pipeline to *de novo* variant calling as <1% for mutations of up to 10 repeat copies (Mitra, et al. 2021; PMID:33442040). Additionally, we now include a more transparent review of GangSTR’s error rate with respect to certain TR motifs in the limitations section (page 11 paragraph 2).

Reviewer Comment 2: The methods for the PheWAS analysis are difficult to follow. The authors mention 148,607 unrelated UKBB participants, and linear/logistic regression on the outcomes. Was this analysis conducted on the 30 probands with *de novo* STR calls relative to all other participants? How did the authors account for what must have been extremely small cell counts? Did the authors include the sequenced parents of the proband in the analysis, or account for relatedness? The authors state that the length-sum was normalized for covariates - why was the phenotype not normalized for these covariates?

Author Response: We now include an overview figure in our submission to clarify this point (please refer to Figure 1). We first identified loci that have *de novo* mutations in family samples. Recognizing the poor statistical power of the family data for performing association with any phenotype, we performed PheWAS of the loci that had *de novo* mutations. We do not perform a PheWAS of the *de novo* mutations themselves. The PheWAS data set excluded all members of the family trios.

We chose to normalize the TR length sum using covariate data instead of normalizing the phenotype because many of the TR length sums were heavily skewed due to variable allele frequencies. Furthermore, not normalizing the phenotypes permitted a more direct comparison between the linear model output and the Cohen’s *d* estimates reported, for example, in Figure 5.

Reviewer Comment 4: In examining the supplemental tables, the authors appear to have conducted PheWAS association tests across the full sample set, which raises several other questions - if there are thousands of people with Notch4 repeats, why are the authors calling this a *de novo* mutation? Are the authors considering the NUMBER of repeats to be a “*de novo*” allele? If so, won't all participants have “*de novo*” repeats SOMEWHERE in the genome?

Author Response: We apologize for the confusing nomenclature here. NOTCH4 contains a *de novo* variant in our family-based analysis. The NOTCH4 mutation in the family trios is an allele within the observed allele spread at the population level. We took all loci containing *de novo* variants and performed a PheWAS in the population-based sample which captures the normal variation at the locus. We hope the addition of Figure 1 helps clarify the analysis approach. The family trios are underpowered to detect relationships between TR length sum and the phenotypes tested. However, we investigated these TRs because they may be hotspots for *de novo* mutational events and for this reason they could have a strong impact on the phenotypic expression in the general population (Acuna-Hidalgo, et al. 2016, PMID: 27894357).

Reviewer Comment 5: In sum, this work has tremendous potential and is straightforward in its goals, but it is difficult to follow and have confidence in the results as presented.

Author Response: We appreciate the Reviewer's encouraging comments. With the addition of a clearer description of quality control metrics, an overview figure to clarify the organization of analyses, and the presentation of structural information as requested by Reviewer 2, we hope the Reviewer is satisfied with the revised content of our submission.

Thank you for your time and I look forward to hearing from you about the review and disposition of our findings.

Sincerely,

Frank R Wendt, PhD

REVIEWERS' COMMENTS

Reviewer #1 (Remarks to the Author):

The authors have extensively revised the manuscript and adequately addressed all of my comments. The manuscript is substantially improved and extended with these various changes and additions.

Reviewer #2 (Remarks to the Author):

The authors have addressed my concerns. I have some very minor additional comments. For their response to my second comment (“given the phewas is based on de novo TRs identified in 39 trios, are they enriching for TR-phenotype associations for traits observed among these trios?”), I would be cautious about the evidence for a greater effect size in the trios for USP30-[CGG]N versus platelet count, as this may be influenced by ‘winner’s curse’. In the abstract, I would recommend also including the corrected P values for the findings presented.

I was also asked to comment on the authors' responses to Reviewer #3. From my understanding of this review, I think the authors have addressed the main concerns. I have just one point to make; in the first comment, the reviewer writes “More information is needed on the quality control, validation, and accuracy of GangSTR calls from whole exome data.”, however, it is not clear to me whether they want this information from the UKB data (i.e, how many de novo variants identified in UKB were validated), or from the original GangSTR paper. The authors response suggests the latter, and if that is what the reviewer intended then the response is adequate. If it is the former, then perhaps the authors have not addressed this comment.

Overall, I do not have any major concerns regarding how the authors have responded to reviewer 3.

19 November 2022

Ref: NCOMMS-22-04483A revised manuscript editorial comments

We would like to thank the Editor and Reviewers for their careful and thorough reading of our manuscript and for their supportive comments. This has been an extremely fruitful review process and we are grateful for the feedback of the Reviewers.

Please find below the response to all the Reviewers' comments. In the main text we highlight in blue all changes made in response to the editorial team requests and the Reviewers' comments.

Reviewer #1 (Remarks to the Author):

The authors have extensively revised the manuscript and adequately addressed all of my comments. The manuscript is substantially improved and extended with these various changes and additions.

Author Response: We thank the Reviewer for their constructive feedback.

Reviewer #2 (Remarks to the Author):

The authors have addressed my concerns. I have some very minor additional comments. For their response to my second comment ("given the phewas is based on de novo TRs identified in 39 trios, are they enriching for TR-phenotype associations for traits observed among these trios?"), I would be cautious about the evidence for a greater effect size in the trios for USP30-[CGG]N versus platelet count, as this may be influenced by 'winner's curse'. In the abstract, I would recommend also including the corrected P values for the findings presented.

Author Response: We now include all FDR-adjusted p-values in the abstract. We also not caution readers to the potential for winner's curse for the USP30-CGG finding in the trio sample.

I was also asked to comment on the authors' responses to Reviewer #3. From my understanding of this review, I think the authors have addressed the main concerns. I have just one point to make; in the first comment, the reviewer writes "More information is needed on the quality control, validation, and accuracy of GangSTR calls from whole exome data.", however, it is not clear to me whether they want this information from the UKB data (i.e, how many de novo variants identified in UKB were validated), or from the original GangSTR paper. The authors response suggests the latter, and if that is what the reviewer intended then the response is adequate. If it is the former, then perhaps the authors have not addressed this comment.

Author Response: We thank Reviewer 2 for taking the extra time to evaluate our responses to Reviewer 3. We interpreted the earlier Reviewer comment as a request for methodological transparency and reliability as Reviewer 3 pointed us to a study showing error rates as high as 2.7% when using GangSTR. The authors of that study transparently report that the high error rate arose from applying user-specified thresholds rather than developer-thresholds. When appropriate developer thresholds were applied, the GangSTR error rate dropped to 0.53% and had the most conservative call rate of the methods tested. Our submission provides detailed description of all sequence-level quality control applied to the UKB (read depth, flanking reads required, etc.) and highlight the appropriate application of GangSTR parameters such that our error rate approximates those reported in prior studies (0.53%).

Thank you for your time and I look forward to hearing from you about the review and disposition of our findings.

Sincerely,

Frank R Wendt, PhD